# Orally delivered MK-4482 inhibits SARS-CoV-2 replication in the Syrian hamster model

Kyle Rosenke[1], Frederick Hansen[1], Benjamin Schwarz[2], Friederike Feldmann[3], Elaine Haddock[1], Rebecca Rosenke[3], Kent Barbian[4], Kimberly Meade-White[1], Atsushi Okumura[1], Shanna Leventhal[1], David W. Hawman[1], Emily Ricotta [5], Catharine M. Bosio[2], Craig Martens[4], Greg Saturday[3], Heinz Feldmann [1✉] & Michael A. Jarvis [1,6,7✉]

The COVID-19 pandemic progresses unabated in many regions of the world. An effective antiviral against SARS-CoV-2 that could be administered orally for use following high-risk exposure would be of substantial benefit in controlling the COVID-19 pandemic. Herein, we show that MK-4482, an orally administered nucleoside analog, inhibits SARS-CoV-2 replication in the Syrian hamster model. The inhibitory effect of MK-4482 on SARS-CoV-2 replication is observed in animals when the drug is administered either beginning 12 h before or 12 h following infection in a high-risk exposure model. These data support the potential utility of MK-4482 to control SARS-CoV-2 infection in humans following high-risk exposure as well as for treatment of COVID-19 patients.

[1] Laboratory of Virology, National Institute of Allergy and Infectious Diseases, National Institutes of Health, Hamilton, MT, USA. [2] Laboratory of Bacteriology, National Institute of Allergy and Infectious Diseases, National Institutes of Health, Hamilton, MT, USA. [3] Rocky Mountain Veterinary Branch, National Institute of Allergy and Infectious Diseases, National Institutes of Health, Hamilton, MT, USA. [4] Research Technologies Branch, National Institute of Allergy and Infectious Diseases, National Institutes of Health, Hamilton, MT, USA. [5] Laboratory of Clinical Immunology and Microbiology, Division of Intramural Research, National Institute of Allergy and Infectious Diseases, National Institutes of Health, Bethesda, MD, USA. [6] University of Plymouth, Plymouth, Devon, UK. [7] The Vaccine Group Ltd, Plymouth, Devon, UK. ✉email: feldmannh@niaid.nih.gov; michael.jarvis@plymouth.ac.uk

Severe acute respiratory syndrome coronavirus 2 (SARS-CoV-2) is the causative agent of coronavirus disease 2019 (COVID-19)[1]. Following the emergence of the virus in late 2019[2], COVID-19 was declared a pandemic by the World Health Organization (WHO) on 11 March 2020[3]. As of late December 2020, there are over 79 million confirmed cases and more than 1.7 million deaths from COVID-19 worldwide[3]. Myriad differences in governmental public health responses, politicization of the pandemic response and societal acceptance of control measures have resulted in differing levels of success in controlling the initial wave of infection around the world[4–7]. Even in those countries that have achieved a higher degree of control of the initial pandemic wave, the unavoidable need to relax highly stringent public health measures has resulted in a rebound of SARS-CoV-2 infections, with a second wave already hitting many countries in the Northern hemisphere[8].

Currently, there are no drugs suitable for high-risk exposure use against SARS-CoV-2. The nucleoside analog, GS-5734 (remdesivir), a non-obligate RNA chain terminator, has been granted emergency use authorization (EUA) by the FDA for the treatment of COVID-19 patients[9]. This EUA was based on the demonstration of a decreased time to recovery in patients hospitalized for severe COVID-19, and was recently expanded to include all hospitalized adult and pediatric patients, irrespective of disease severity[9,10]. In preclinical animal studies, which are more amenable than clinical trials for assessment against high-risk exposure, GS-5734 administered 12 h after SARS-CoV-2 infection was shown to lower lung viral load and lung pathology, although treatment had no effect on shedding from the upper respiratory tract[11]. The use of GS-5734 for control of disease in symptomatic COVID-19 patients remains a point of contention[12]. Currently, GS-5734 can be administered only via the intravenous route, which makes its application to the control of high-risk exposure challenging.

MK-4482 (known previously as EIDD-2801) is an orally administered bioavailable prodrug (5′-isobutyric ester form) of the cytidine nucleoside analog EIDD-1931 ($\beta$-D-N[4]-hydroxycytidine; NHC)[13]. Using a high throughput screen of nucleoside analogs, EIDD-1931, the active compound resulting from hydrolysis of MK-4482, was identified as a broad activity inhibitor of influenza A and respiratory syncytial viruses, with initial functional assays showing the drug to function primarily as an RNA mutagen rather than chain terminator[14]. Originally developed for the treatment of hepatitis C virus (HCV) in early the 2000s[15], recent studies indicated potent activity of EIDD-1931 against SARS-CoV-2 in multiple cell types, including biologically relevant epithelial cells in vitro, and against MERS-CoV-1 and SARS-CoV-1 coronaviruses in mouse models when administered shortly before as well as following infection[16].

In this work, we determine the half-maximal inhibitory concentration (IC$_{50}$) value for EIDD-1931 in tissue culture and subsequently assess the potential of MK-4482 following oral administration to control SARS-CoV-2 in the highly susceptible Syrian hamster model[17,18]. We show that MK-4482, when administered either starting at 12 h prior to SARS-CoV-2 infection, or even 12 h post-infection, significantly decreases viral lung loads and pathology, but does not affect shedding from the upper respiratory tract. These findings support the potential of MK-4482 as an orally administered drug for high-risk exposure and possibly therapeutic use in humans.

## Results

First, we determined the in vitro inhibitory effect of EIDD-1931 on SARS-CoV-2 replication in Calu-3 cells, a disease-relevant human lung epithelial cell line. Cells were pretreated with differing drug concentrations and the effect on viral RNA load in tissue culture supernatant was determined at 24 h after infection by quantitative reverse transcriptase polymerase chain reaction (RT-PCR) (Fig. 1a). EIDD-1931 treatment resulted in a decrease in SARS-CoV-2 replication by approximately 3-logs (880-fold) when compared to no drug controls (Figs. 1a, b). The half-maximal inhibitory concentration (IC$_{50}$) value for EIDD-1931 was shown to be at sub-micromolar levels in Calu-3 cells at 414.6 nM (Fig. 1c). Viability was also assessed over the differing concentrations, demonstrating only minimal cellular toxicity at the highest drug concentration (Fig. 1d).

Having verified in vitro efficacy and determined the IC$_{50}$ value of EIDD-1931, we next assessed the efficacy of the MK-4482 prodrug in the Syrian hamster model, which is regarded as a preclinical model of mild disease, with animals having self-limiting pneumonia[17,18]. Given the possibility for oral dosing, we were interested in the utility of MK-4482 as a treatment following high-risk exposure. The Syrian hamster model used for these studies is a recently established model that has further expanded understanding of key infection parameters absent from initial iterations of this preclinical model[18]. Two groups of hamsters ($n = 6$ per group) were treated with MK-4482 (250 mg/kg) by oral gavage 12 h and 2 h before (pre-infection treatment group) or 12 h post-infection (post-infection treatment group). This dosing regimen was based on previous studies using MK-4482 in preclinical rodent models of SARS-CoV-1 and MERS[16]. Animals were then dosed every 12 h with MK-4482 (250 mg/kg). A control group was treated using the same route and timing as the pre-infection group with vehicle only (see schematic; Fig. 2a). Hamsters were infected intranasally with SARS-CoV-2 using a dose of $5 \times 10^2$ TCID$_{50}$ (100 times infectious dose 50; ID$_{50}$). The ID$_{50}$ value was determined in a separate study concerned with further refinement of the Syrian hamster SARS-CoV-2 model[18]. Treatment in all groups was continued for 3 consecutive days and hamsters in all groups were euthanized on day 4 post-infection at the peak of virus replication[18].

Disease in Syrian hamsters following SARS-CoV-2 infection is transient, peaking at or around 4 days post-infection with minimal clinical signs[17–20]. Consistent with this observation, no substantial clinical symptoms were observed in any group over the course of the study, including an absence of any discernible differences in weak weight loss (Supplementary Figure 1). Virus shedding was measured with oral swabs collected on day 2 and 4 post-infection. Levels of viral RNA in the oral cavity decreased from day 2 to 4, but were similar between all groups at these two time points of analysis (approximately $10^8$ and $10^7$, for day 2 and 4 post-infection, respectively) (Fig. 2b). Although more variable, infectious titers recovered from oral swabs were consistent with the genome copy data in that similar levels of infectious virus were detected in all groups at each time point (Fig. 2c). Lung tissue samples were collected at the peak of virus replication and disease, day 4 post-infection, for analysis. In contrast to levels of shedding, a 1-log decrease in viral RNA was detected in the lungs of pre-infection and post-infection groups, respectively, when compared to the vehicle control group (Fig. 2d). This corresponded to a 2-log decrease in infectious virus in the lungs of the MK-4482 treated groups when compared to the vehicle controls (Fig. 2e).

Lung samples were taken for histopathological analyses, and results are shown in Fig. 3a–f. Analysis revealed pulmonary lesions consisting of a moderate-marked broncho-interstitial pneumonia centered on terminal bronchioles and extending into the adjacent alveoli. Multifocal necrotic epithelial cells and moderate numbers of infiltrating neutrophils and macrophages with abundant luminal cellular exudate in the bronchi and bronchioles were also present. Alveolar septa were expanded by

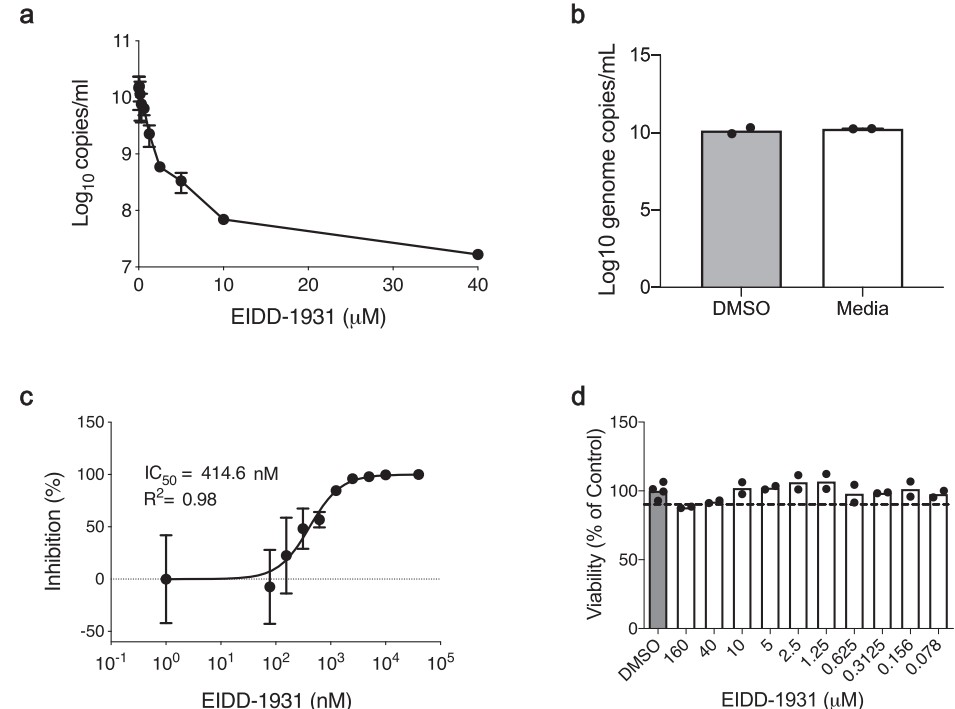

**Fig. 1 EIDD-1931 inhibits SARS-CoV-2 replication in human lung epithelial Calu-3 cells.** Cells were pretreated for 1 h with differing EIDD-1931 concentrations, followed by infection with SARS-CoV-2 at a MOI of 0.01 for 1 h. After 1 h, media was replaced, and cells were cultured in the presence of drug for 24 h at 37 °C in a 5% $CO_2$ incubator. **a** Virus yield in the cell supernatant was measured by quantitative RT-PCR of clarified culture supernatant by using primer and probe sets to quantify total viral RNA (N gene; genomic and subgenomic RNA). **b** DMSO control versus media. Replicates were analyzed in duplicate (mean ± SD are shown). **c** $IC_{50}$ values were determined using results from the RT-PCR following log-based transformation of drug concentrations and normalization to percentage inhibition based on diluent alone controls by fitting to drug-dose response curves using Prism software. **d** Absence of toxicity (>90% viability; shown by dotted line) at highest EIDD-1931 concentration used for analysis of SARS-CoV-2 replication (40 μM) was confirmed using CellTiter-Glo® 2.0 Assay (Promega, Corp., Madison, WI, USA) as per the manufacturer's protocol. Individual sample values of replicates are shown.

edema fluid and leukocytes. Moderate type II pneumocyte hyperplasia was noted in more consolidated areas with abundant alveolar macrophages, cellular exudate, and edema. Blood vessels were surrounded by moderate numbers of lymphocytes that multifocally aggregated in vascular tunics and elevated the overlying epithelium. Low numbers of syncytial cells were noted in the bronchioles and alveoli. These described lesions affected 20–50% of pulmonary tissue in the vehicle control groups and while the pre-infection and post-infection treatment groups had similar lesions, they were noticeably less abundant compared to the vehicle control. One animal in each of the pre- and post-infection treatment groups had no lesions at all. Pneumonia in the remaining animals affected roughly 5–15% of the lung tissue, but lesions were minimal to mild.

Immunoreactivity against SARS-COV-2 antigen was used to further compare the lung samples between the three different treatment groups (Fig. 3g–i). Antigen staining was observed in the bronchial and bronchiolar epithelium, type I and II pneumocytes as well as a small number of pulmonary macrophages. A positive pixel analysis on whole lung slides demonstrated a significant difference in viral antigen present among the three groups. The total number of positive pixels was divided by the area of lung scanned to determine a percentage of lung containing viral antigen. This analysis revealed that the vehicle controls contained significantly more antigen than the treated groups, with the vehicle controls having on average 4.71 times more antigen signal than pre-infection treatment animals and 3.68 times more signal than post-infection treatment animals. Post-infection treatment animals exhibited a slightly higher antigen signal

than pre-infection treatment animals, but the difference was not significant (Fig. 4a).

To evaluate the pharmacokinetics of MK-4482 in the lungs of animals, MK-4482 and the EIDD-1931 metabolite were measured in clarified lung homogenate by liquid chromatography and mass spectrometry (LCMS) at the point of necropsy. Since SARS-CoV-2 is a respiratory disease, levels of drug in lung tissue are expected to be the best indicator of therapeutic potential. All treated animals displayed detectable levels of EIDD-1931 in the lung and levels were similar across treatment groups (pre-infection: 18.80 ± 5.97 nmol/$g_{lung}$, post-infection 17.56 ± 5.49 nmol/$g_{lung}$) (Supplementary Table 1) (Fig. 4b). In line with its demonstrated rapid hydrolysis to EIDD-1931 following absorption, MK-4482 was not detected in the tissue[13,16]. Concentration is difficult to estimate in tissues due to non-homogenous drug distribution and organ hydration. On average, water content of the lung is approximately 80% by weight and this number can be used to calculate a conservative estimated EIDD-1931 concentration in the tissue under the assumptions of homogenous distribution and hydration[21]. These estimates suggest a concentration of 15.04 ± 4.78 μM in the pre-infection group and 14.05 ± 4.39 μM in the post-infection group at the point of necropsy (12 h post-final MK-4482 dose) (Supplementary Table 1) (Fig. 4b). These values compare well with previous studies where a single oral dose of MK-4482 at 128 mg/kg in ferrets (compared to 250 mg/kg in our study) resulted in EIDD-1931 lung concentrations of 10.7 ± 1.2nmol/g[13]. While our study was not designed to assess detailed pharmacokinetics of MK-4482, the similarity in EIDD-1931 levels observed in the lungs of hamsters with those of ferrets from this

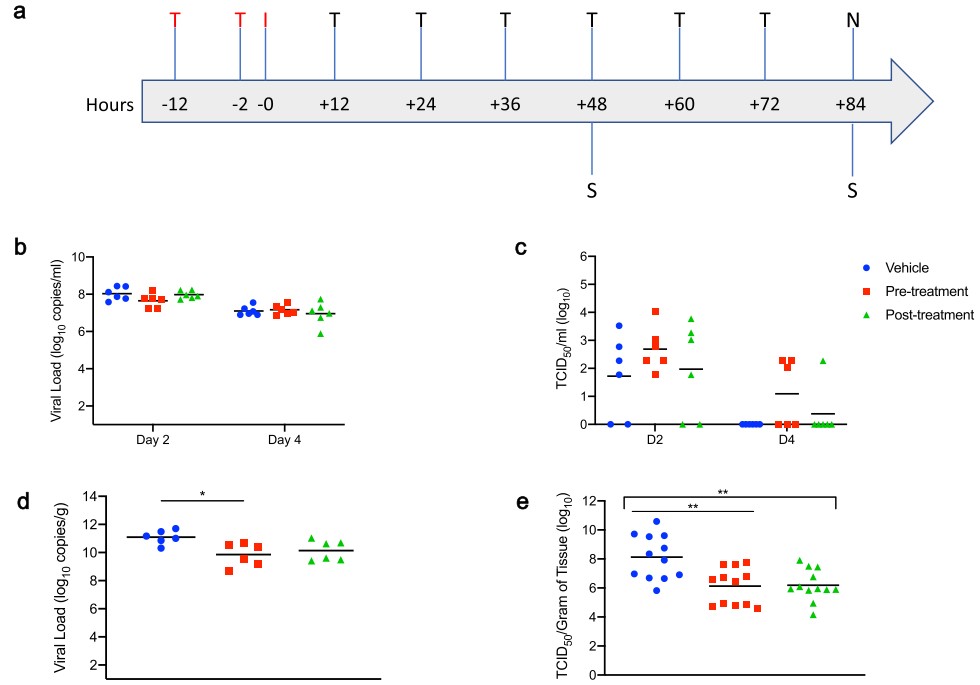

**Fig. 2 Syrian hamster model—study design, viral shedding, viral load, infectious titers, and viral antigen. a** Hamsters were intranasally infected with SARS-CoV-2. MK-4482 was administered pre-infection at 12 and 2 h before infection, or post-infection starting 12 h post-infection. Treatment was continued in both groups every 12 h for 3 consecutive days. Animals were euthanized on day 4 and lungs were harvested. T = treatment (red: pre-infection and black: post-infection treatments); I = infection; S = swab samples and N = necropsy. **b** Oral swab samples ($N = 6$ per group) were collected on days 2 and 4 post-infection and viral shedding determined by RT-PCR ($p$ Value Vehicle vs Pre-treatment = >0.9999, $p$ Value Vehicle vs Post-treatment = >0.9999, One-way ANOVA, Kruskal–Wallis test). **c** Oral swab samples ($N = 6$ per group) were titered for infectious virus ($TCID_{50}$) on Vero E6 cells[43] ($p$ Value Vehicle vs Pre-treatment = 0.5701, $p$ Value Vehicle vs Post-treatment = >0.9999, One-way ANOVA, Kruskal–Wallis test). **d** Lung viral loads ($N = 6$ per group) were determined by using RT-PCR ($p$ Value Vehicle vs Pre-treatment = 0.0189, $p$ Value Vehicle vs Post-treatment = 0.1032, One-way ANOVA, Kruskal–Wallis test). **e** Lung samples ($N = 6$ per group) were homogenized and titered for infectious virus ($TCID_{50}$)[43] on Vero E6 cells. Two independent lung samples were measured from each animal ($N = 12$ per group) ($p$ Value Vehicle vs Pre-treatment = 0.0091, $p$ Value Vehicle vs Post-treatment = 0.0102, One-way ANOVA, Kruskal–Wallis test). **b–e** Blue circle, vehicle control; red square, pre-infection treatment; green triangle, post-infection treatment. Summary of Results: **b, c** No statistical significance in virus shedding (RT-PCR or $TCID_{50}$) between either of the two MK-4482 treatment groups and vehicle controls. **d** Significant difference in lung viral loads (RT-PCR) between pre-infection group compared to the vehicle control. Although post-infection group trended towards lower levels, no significant difference between this group and vehicle control. **e** Infectious titers in the lungs ($TCID_{50}$) were significantly different between both pre-infection and post-infection groups, compared to vehicle control group, but no significance was found between treatment groups from each other. One-way ANOVA followed by Kruskal–Wallis analysis and a pairwise Wilcox test was used to analyze differences among groups. *$p < 0.05$, **$p < 0.008$.

earlier study[13] suggests a comparable drug behavior across these models.

MK-4482 has been shown to function as an RNA mutagen[13,14,16]. Viral RNA isolated from lung samples was sequenced and examined for mutations. When compared to the vehicle, viral genomes from MK-4482 treated animals had a significant accumulation of nucleotide substitutions (Supplementary Table 2). The mutational spectra associated with this cytosine analog was also consistent with its function, substituting as either a cytosine or uracil residue, which resulted in increased accumulation of adenosine-to-guanosine and cytosine-to-uracil transitions in viral genomes (Supplementary Table 2). Together, these results are consistent with the RNA mutagenesis function of MK-4482 in the reduction of infectious virus and disease in treated animals.

## Discussion

In the present study, we used the established Syrian hamster animal model[17,18] to assess the inhibitory effect of the nucleoside analog MK-4482 on SARS-CoV-2 replication in vivo. Our study shows the capacity of MK-4482 to substantially reduce the replication of SARS-CoV-2 in the lungs based on both viral RNA

genome copy number and levels of infectious virus. Importantly, this control of virus replication was associated with markedly reduced lung pathology. MK-4482 has been shown to inhibit the replication of other related human coronaviruses, MERS-CoV and SARS-CoV-1 in mouse models[16].

Following submission of our manuscript, a preprint of a similar study in a Syrian hamster model was released[22]. This study also showed an inhibitory effect of MK-4482 on SARS-CoV-2 replication and lung disease. Although similar in design, a number of details including SARS-CoV-2 challenge titer and virus strain differ between the two studies, and may account for the greater reported effect of MK-4482 pre-treatment on virus lung levels in the later study. Importantly, however, both studies support a significant effect of MK-4482 pre-treatment on SARS-CoV-2 replication in the lung, both in terms of viral RNA and virus titer, which was also reflected in decreased lung pathology. In our study, treatment at 12 h post-infection maintained the effect of MK-4482 on both viral RNA (1-log) and virus lung titers (2-logs), which was also seen as decreased SARS-CoV-2 antigen lung levels and by its protective effect against lung pathology. In contrast, an inhibitory effect was largely lost in the preprint study when treatment was initiated at 24 h post-infection (the 12 h post-infection time point was not tested)[22]. Whether these reflect real

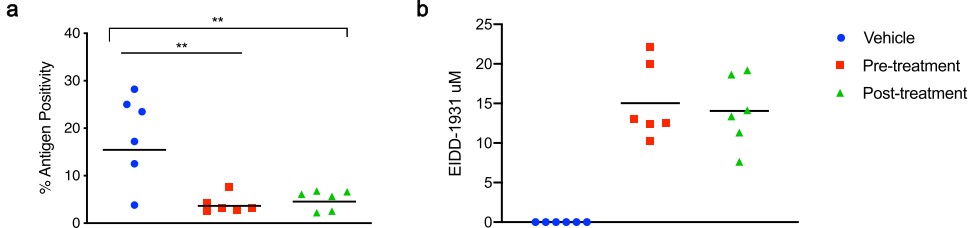

**Fig. 3 Pathological analysis of the lung tissue.** Hematoxylin and eosin (H&E) staining was used on lung sections to examine lung pathology post-inoculation. Immunohistochemistry (IHC) was used to detect viral antigen in the same lung sections from each animal ($N = 6$ per group). **a, d, g** Untreated vehicle control, (**b, e, h**) pre-infection treatment with antiviral drug MK-4482 and (**c, f, i**) post-infection treatment with MK-4482. (**a–f**) H&E stain (**g, h, i**) IHC for SARS-CoV-2 nucleocapsid antibody. **a** Lung 20X: multifocal, moderate broncho-interstitial pneumonia. **b, c** Lung 20X: minimal peribronchial interstitial pneumonia. **d** Lung 200X epithelial cell necrosis (arrow), edema (asterisk), interstitial pneumonia (arrowhead). **e, f** peribronchial and interstitial infiltrates (arrow). **g** Lung 20X; insert 200X: numerous immunoreactive bronchiolar epithelial cells, type I and II pneumocytes and fewer macrophages. **h, i** Lung 20X; insert 200X: scattered to moderate numbers of immunoreactive bronchiolar epithelial cells, type I and II pneumocytes and macrophages. Pictures were taken in RGB color space (sRGB IEC61966-2.1) with a threshold set to 128. **a–c, g–i** Scale bar is 200μm. **d–f** Scale bar is 20μm.

**Fig. 4 Morphometric analysis of viral antigen and drug concentration in the lungs. a** A longitudinal cross section of the right lung of each animal ($N = 6$ per group) was stained for viral antigen and scanned to measure the total amount of viral antigen present in the lung section. **b** EIDD-1931 concentrations in the lungs. **a, b** Blue circle, vehicle control; red square, pre-infection treatment; green triangle, post-infection treatment. Summary of results. **a** The area of lung staining positive for viral antigen showed a statistically significant difference between both of the MK-4482 treatment groups, compared to vehicle controls. No difference between individual treatment groups was present. (One-way ANOVA followed by Kruskal–Wallis analysis and a pairwise Wilcox test was used to analyze differences among groups. **p < 0.008).

differences in the potential treatment window for MK-4482 between the two studies is not possible to ascertain without additional investigation, but is consistent with direct acting antivirals being most effective in modifying disease outcome when administered as early as possible following infection[23].

Since our initial submission, a study by Cox et al (2021)[24] using a ferret SARS-CoV-2 transmission model has shown an inhibitory impact of MK-4482 on virus levels in the upper respiratory tract corresponding to a decreased virus transmission between cohoused animals. The ferret model appears to be more suited to SARS-CoV-2 transmission studies rather than the assessment of lung load and associated lung disease, with an absence of detectable titers of virus in lung tissues of ferrets at any time post-infection. The results from the ferret study appear to contradict our observed absence of any differences in viral RNA levels of oral swabs between treatment groups and controls. Hamsters have been used to assess the effect of drug treatments on transmission of SARS-CoV-2 between cohoused hamsters[25]. Although the results from the ferret model strongly support an inhibitory effect of the drug on transmission[24], additional studies in the hamster will be needed to assess the effect of MK-4482 on this parameter.

Currently, only a single drug (GS-5734) has been given EUA by the FDA for treatment of SARS-CoV-2 induced COVID-19 disease[9], with broader adoption of this drug for the treatment of COVID-19 patients remaining an area of ongoing discussion[12]. Rather than having an impact on mortality, the FDA EUA was based on a demonstration of reduced recovery time for hospitalized patients with COVID-19[10]. In a study performed in the rhesus macaque model, GS-5734 administered at 12 h post-infection was shown to lower both the peak infectious titers of SARS-CoV-2 in bronchoalveolar lavage (BAL) and virus genome copy number in the lung at day 7 post-infection by approximately 2-logs[11]. Currently, there is no data showing the efficacy of GS-5734 against SARS-CoV-2 in the Syrian hamster model, but treatment starting a day prior to infection and continued twice daily thereafter resulted in significant improvement of SARS-CoV-2 infection in adenovirus 5-hACE2 transduced mice[26]. However, the hamster and macaque models appear relatively comparable, with both being associated with a rapid increase in SARS-CoV-2 replication in the lung and other respiratory tissues and rather mild clinical disease[17,18,27]. Given these similarities, MK-4482 should likely be considered as a potential additional treatment option for COVID-19 patients.

Similar to GS-5734, MK-4482 exhibits broad inhibition of divergent RNA viruses[13–16,28–30]. Although both drugs are nucleoside analogs, MK-4482 has been shown to function as an RNA mutagen inducing genome catastrophe[16,31], while GS-5734 is a non-obligate RNA chain terminator[32]. The function of MK-4482 as an RNA mutagen may raise concerns regarding 'off-target' mutagenic toxicity. However, even at an EIDD-1931 dose of 500 mg/kg, treatment of mice in a MERS-CoV model did not increase mutation rates of the ISG15 mRNA transcript, a gene highly induced during MERS-CoV infection, whilst viral genes accumulated mutations[16]. The incorporation of ribonucleosides has also been shown to be highly selective for RNA compared to host DNA[33]. For example, the guanosine ribonucleoside analog, ribavirin, which has several mechanisms of action including one of RNA mutation/error catastrophe, has been used in the past in patients, including children with severe lower respiratory tract infections and is still used in combination for the treatment of hepatitis C[34,35]. If deemed safe, MK-4482 would join GS-5734 as the second broadly direct acting antiviral to target emerging RNA viruses, and in this case, specifically SARS-CoV-2.

Infectious disease pathology is a complex interplay between the pathogen and the host. Consequently, strategically planned combination therapy may be more effective than the use of single drugs. Combinations of drugs with different mechanisms of action would be preferable. Such combination therapy has been shown to be highly effective for the control of other viral pathogens, notably human immunodeficiency and hepatitis C virus infection[36,37]. Therefore, the combination of MK-4482, an RNA mutagen, with the non-obligate RNA chain terminator, GS-5734, may yield additional efficacy in the treatment of SARS-CoV-2 infections. Additional combination partners could be potent neutralizing antibodies[38]. In addition, immune response modifying drugs such as dexamethasone have been shown to be effective for the later deleterious host responses associated with COVID-19 disease[39]. The combination of such a therapeutic with direct antivirals, such as MK-4482 and GS-5734, may increase treatment efficacy, especially in more severe cases. Any potential combination would need to be tested in vitro and in preclinical models to establish the anticipated synergy or added effect, rather than any unexpected absence or antagonism between drug combinations.

GS-5734 is currently only administered by the intravenous route. A clear advantage of MK-4482 is the capacity for oral administration, which opens up the possibility for use of the drug as a post-exposure treatment. Our data suggest that initiation of treatment within 12 h of a productive exposure resulting in infection significantly reduces SARS-CoV-2 replication and associated pathology in the lung target organ. Consistent with this idea, direct acting antivirals, including GS-5734 have been shown to be most effective in modifying disease outcome when administered early following infection[23]. If adequately priced for widespread global use, we believe that MK-4482 should be considered as an oral post-exposure application for SARS-CoV-2.

## Methods

**Biosafety and ethics**. Work with infectious SARS-CoV-2 complied with all relevant ethical regulations for animal testing and research. The hamster study received ethical approval from the Rocky Mountain Laboratories Animal Care and Use Committee (IACUC, Protocol # 2020-044-E) and was performed in a high biocontainment laboratory at Rocky Mountain Laboratories (RML), NIAID, NIH. Animal work was performed by certified staff in an Association for Assessment and Accreditation of Laboratory Animal Care International accredited facility. Work followed the institution's guidelines for animal use, the guidelines and basic principles in the NIH Guide for the Care and Use of Laboratory Animals, the Animal Welfare Act, United States Department of Agriculture and the United States Public Health Service Policy on Humane Care and Use of Laboratory Animals.

Syrian hamsters were group housed in HEPA-filtered cage systems enriched with nesting material and were provided with commercial chow and water ad libitum. Animals were monitored at least twice daily throughout the study.

**Virus and cells**. SARS-CoV-2 isolate nCoV-WA1-2020 (MN985325.1) (https://www.ncbi.nlm.nih.gov/nuccore/MN985325) was kindly provided by the Centers for Disease Control and Prevention, Atlanta, GA, USA[40] and propagated once at RML in Vero E6 cells in high glucose DMEM (Sigma) supplemented with 2% fetal bovine serum (Gibco), 1 mM L-glutamine (Gibco), 50 U/ml penicillin and 50 μg/ml streptomycin (Gibco). The virus stock used was free of contaminations and confirmed to be identical to the initial deposited Genbank sequence (MN985325.1). Vero E6 cells were maintained in high glucose DMEM supplemented with 10% fetal calf serum, 1 mM L-glutamine, 50 U/mL penicillin and 50 μg/mL streptomycin.

**Syrian hamster study design**. Hamsters were divided into groups for either pre-infection or post-infection MK-4482 treatment ($n = 6$ per group). Groups were then treated with MK-4482 (250 mg/kg) [MedChemExpress dissolved in 10 % polyethylene glycol (PEG)−400; 2.5% Cremophor RH40 in water] at 12 h and 2 h prior to infection (pre-infection group) or 12 h following infection (post-infection group). Treatment was then maintained with 12 h dosing until the completion of the study 84 h post-infection (day 4). A third group consisted of vehicle control animals that received the same dosing schedule and volume as the pre-infection group. All groups were infected intranasally with $5 \times 10^2$ TCID$_{50}$ of SARS-CoV-2 (25 μL/nare). Animal weights were collected once daily and animals were monitored twice daily for disease signs and progression. All procedures were performed on anesthetized animals. Oral swabs were collected on days 2 and 4 post-infection.

Animals were euthanized on day 4 post-infection and lung tissues were collected at necropsy for pathology and virology.

**Liquid chromatography and mass spectrometry (LCMS)**. LCMS grade water, methanol, acetonitrile and acetic acid were purchased through Fisher Scientific. All synthetic standards for molecular analysis were purchased from MedChemExpress. Clarified lung homogenates were gamma-irradiated (2 megarads) for removal from biocontainment according to IBC-approved protocol[41]. Standard curves of MK-4482 and EIDD-1931 were made in lung homogenate from uninfected animals and subjected to irradiation to account for molecular degradation. Samples were prepared for analysis by adding 300 μL of methanol to 100 μL of homogenate and incubating at 4 ºC for 30 min to precipitate macromolecules. Samples were centrifuged at 16,000g at 4 ºC and the supernatant was transferred to a sample vial for LCMS analysis. Samples were separated by HILIC chromatography on a Sciex ExionLC™ AC system. Samples were injected onto a Waters XBridge® Amide column (130 Å, 3.5 μm, 3 mm × 100 mm) and eluted using a binary gradient from 95% acetonitrile, 0.8% acetic acid, 10 mM ammonium acetate to 50% acetonitrile, 0.8% acetic acid, 10 mM ammonium acetate over 8 min. Analytes were measured using a Sciex 5500 QTRAP® mass spectrometer in positive mode with electrospray ionization (CUR: 40, CAD: Med, ISV: 2500, Temp: 450, GS1: 50, GS2: 50). Multiple reaction monitoring (MRM) was performed using the optimized conditions in Supplementary Table 3. To ensure signal fidelity triggered spectra were compared back to synthetic standards. Previously published MRM signals for biological nucleosides were utilized to confirm minimal interference at the retention time of interest[42]. All analytes were quantified against an 8-point calibration curve of the respective synthetic standard prepared in the target matrix and processed in the same manner as experimental samples. Limit of quantification in lung homogenate after irradiation was 5 ng/mL for EIDD-1931 and 50 pg/mL for MK-4482.

**Virus load**. RNA was extracted from swabs using the QIAamp Viral RNA kit (Qiagen) according to the manufacturer's instructions. Tissues were homogenized in RLT buffer and RNA was extracted using the RNeasy kit (Qiagen) according to the manufacturer's instructions. For detection of viral RNA, 5 μl RNA was used in a one-step real-time RT-PCR against the N gene which detects genomic and subgenomic RNA[17] using the Rotor-Gene probe kit (Qiagen) according to instructions of the manufacturer. In each run, standard dilutions of RNA standards counted by droplet digital PCR were run in parallel, to calculate copy numbers in the samples. A complete list of primers is shown in Supplementary Table 4.

**Virus titration**. Virus isolation was performed on lung tissues by homogenizing the tissue in 1 mL DMEM using a TissueLyser (Qiagen) and inoculating Vero E6 cells in a 96-well plate with 200 μL of a 1:10 dilution series of the cleared homogenate. One hour after inoculation of cells, the inoculum was removed and replaced with 200 μL DMEM (Sigma-Aldrich) supplemented with 2% fetal bovine serum, 1 mM ʟ-glutamine, 50 U/mL penicillin and 50 μg/mL streptomycin. Six days after inoculation, cytopathogenic effect was scored and the TCID$_{50}$ was calculated using the Reed-Muench method[43].

**Histopathology and immunohistochemistry**. Histopathology and immunohistochemistry were performed on hamster lung tissues. Tissues were fixed in 10 % Neutral Buffered Formalin with two changes, for a minimum of 7 days according to IBC-approved SOP. Tissues were placed in cassettes and processed with a Sakura VIP-6 Tissue Tek, on a 12-h automated schedule, using a graded series of ethanol, xylene, and PureAffin. Embedded tissues were sectioned at 5 μm and dried overnight at 42ºC prior to staining. Specific anti-CoV immunoreactivity was detected using Sino Biological Inc. SARS-CoV/SARS-CoV-2 nucleocapsid antibody (Sino Biological cat#40143-MM05) at a 1:1000 dilution. The secondary antibody was the Vector Laboratories ImPress VR anti-mouse IgG polymer (cat# MP-7422). The tissues were then processed for immunohistochemistry using the Discovery Ultra automated stainer (Ventana Medical Systems) with a ChromoMap DAB kit (Roche Tissue Diagnostics cat#760-159). The tissues slides were scanned with the Aperio ScanScope XT (Aperio Technologies, Inc.) and the entire section analyzed with the ImageScope Positive Pixel Count algorithm (version 9.1)[44]. All tissue slides were analyzed by a board-certified veterinary pathologist.

**Next-generation sequencing**. Lung derived RNA samples were treated with Ribo-Zero H/M/R rRNA (Illumina, San Diego, CA) depletion mix following a 40μl total volume; 4μL reaction buffer, 8μL probes, and 28μL sample. All incubation times followed the Ribo-Zero manual. After Ampure RNACleanXP (Beckman Coulter, Brea, CA) purification, the enriched RNA was eluted in 6μL of water. Following the *Truseq Stranded mRNA Library Preparation Guide, Revision E.*, (Illumina, San Diego, CA), 5μL was added to 13μL of Elute-Frag-Prime Buffer and continued through second-strand cDNA. Library preparation continued with the adenylation of ends following the manufacturer's recommendations. Final libraries were visualized on a BioAnalyzer DNA1000 chip (Agilent Technologies, Santa Clara, CA) and quantified using *KAPA Library Quant Kit (Illumina) Universal qPCR Mix* (Kapa Biosystems, Wilmington, MA) on a CFX96 Real-Time System (BioRad, Hercules, CA). Libraries were diluted to 2 nM stock, pooled together in equimolar concentrations and sequenced on an Illumina MiSeq using a Micro v2 sequencing kit at 2 × 150-bp. These data were used to make a new 2 nM pool so that virus coverage was normalized across samples. Subsequent sequencing was performed on an Illumina NextSeq 550 using a Mid Output v2.5 sequencing kit at 2 × 150-bp. Viral genome read depth coverage was greater than 100× for all samples.

**Next-generation sequencing data analysis**. Raw fastq reads were adapter trimmed using Cutadapt v 1.12[45], followed by quality trimming and quality filtering using the FASTX Toolkit (Hannon Lab, CSHL). Reads were paired up and aligned to the SARS-CoV-2 genome from isolate SARS-CoV-2/humanUSA/WA-CDC-WA1/2020 (MN985325.1) using Bowtie2 v 2.2.9[46]. PCR duplicates were removed using Picard MarkDuplicates v 2.18.7 (Broad Institute). Variant detection was performed using GATK HaplotypeCaller v 4.1.2.0[47] with ploidy set to 2.

**Statistical analyses**. Data was collected in Excel, v16.4 and graphed using Prism v8 (GraphPad). Statistical analysis was performed in R version 4.0.2. The difference in viral load between study arms was assessed by One-way ANOVA followed by a Kruskal-Wallis test and a pairwise Wilcoxon rank sum test to correct for multiple comparisons.

**Reporting summary**. Further information on research design is available in the Nature Research Reporting Summary linked to this article.

## Data availability

The data that support the findings of this study are available from the corresponding author upon reasonable request. Next generation sequencing data have been deposited in NCBI under BioProject accession number PRJNA691961. Source data are provided with this paper.

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

## Acknowledgements

The authors are thankful to the animal caretakers and histopathology group of the Rocky Mountain Veterinary Branch (NIAID, NIH) for their support with animal related work, and Anita Mora (Visual and Medical Arts Unit, NIAID, NIH) for help with the display items. This work was funded by the Intramural Research Program of the National Institutes of Allergy and Infectious Diseases (NIAID) and National Institutes of Health (NIH). MAJ is funded through The Vaccine Group Ltd, and the University of Plymouth.

## Author contributions

K.R., F.H., H.F., and M.A.J. contributed to the design, execution and data analysis, and writing of the manuscript. B.S. and C.M.B. contributed to the metabolite analysis and editing. G.S. and R.R. contributed to histological and pathology support and analysis. F.F., E.H., K.M.-W., A.O., S.L., D.W.H. contributed experiment support and data analysis. C.M. and K.B. contributed to the sequencing and mutational analysis. E.R. contributed to data analysis. All authors reviewed and contributed to preparation of the final manuscript.

## Competing interests

The authors declare no competing interests.
