## [Peer Review File · Nature Communications]

Reviewer #1 (Remarks to the Author):

This is a revised and expanded study demonstrating efficacy of MK-4482 against SARS-CoV-2 in the Syrian golden hamster model. Data are clearly presented and important. Previous critiques have been mostly well addressed and the addition of infectious titers (in addition to RNA copy numbers originally shown) confirms efficacy of MK-4482 in reducing lung virus load (approx. 1 log unit of RNA copies and 2 log units of virus titer). It should be added to the discussion, however, that a recently released preprint reported considerably greater efficacy of MK-4482 against SARS-CoV-2 in Syrian golden hamsters, resulting in a reduction of RNA copies by approx. 3 log units and of lung virus titers by 3.3 log units. Since animal model, viral target, dose levels, and treatment regimens of both studies are very similar, please discuss possible reasons for these discrepancies.

In contrast to the clarification of lung virus burden, however, the new data have further deepened the oral swab conundrum that was raised by previous reviewers. It is shown that viral RNA load in oral swabs was very consistently approx. 10^8 copies/ml in all animals on day 2 (fig 2B). Looking at virus load in vehicle-treated animals alone, this translates to anything from undetectable (2 animals) to approx. 3.5×10^3 TCID₅₀ units/ml (1 animal) and anything in between (3 animals; fig 2C). How is that level of variation possible? On day 4, vehicle animals had, again quite consistently, approx. 10^7 RNA copies/ml, but no infectious virus could be detected in swabs of any vehicle animal. In the lung samples, RNA load of approx. 10^{11} copies/g corresponded to approx. 10^8 TCID₅₀/g, which largely matches the ratio of RNA:TCID₅₀ reported for SARS-CoV-2 in other hamster studies and in other animal species. How can it be that this ratio is so strikingly different in the oral swab samples, and how can these data be interpreted? These discrepancies need to be clarified.

Possibly a problem of the model or the sampling technique, the lack of correlation of RNA copies to infectious particles and the brief and inconsistent virus presence in the hamster upper respiratory tract confound interpretation of the effect on MK-4482 on virus shedding. Conclusions regarding the effect of treatment on viral transmission cannot be reached without an actual hamster transmission study. Possible limitations of the model need to be considered. This discussion should also cover the recently published data of efficacy of MK-4482 against SARS-CoV-2 replication and spread in ferrets, since in ferrets MK-4482 reduced virus burden in the upper respiratory tract and blocked virus transmission.

In their response to reviews the authors reference as supporting evidence that remdesivir blocked virus load in the lower, but not upper, respiratory tract in macaques. That is correct, but one needs to be careful to base conclusions about efficacy of one drug on that of an unrelated other. Remdesivir by intention (originally developed against HCV) and chemical design shows highest tissue exposure of the active form in liver, opposite to the tissue distribution of the active anabolite of MK-4482, and exposure of the active form of MK-4482 in the upper respiratory tract of hamsters is currently unknown.

In lines 179-180, please update your statement that this “is the first demonstration of inhibition of SARS-CoV-2 [...] in any animal model” by MK-4482, since there are published MK-4482 ferret data, as well as preprint releases covering use of MK-4482 in hamsters and mice.

The implied promise of synergistic effects of combination therapy with two nucleoside analogs, one an RNA mutagen and the other a chain terminator (lines 214-216), may be problematic, since in reality the latter drug undermines the MoA of the former.

Reviewer #3 (Remarks to the Author):

This study by Rosenke et al. examines the use of MK-4482 as a potential antiviral drug for SARS-CoV-2. Currently, remdesivir is the only drug to receive approval for COVID-19 treatment, but its intravenous delivery route is not ideal. MK-4482 is an oral prodrug which gets metabolised into a nucleoside analogue, EID-1931.

This version of the manuscript is substantially improved, particularly in these 3 areas:

- 1) There is now infectious titer data shown for both oral and lung samples in Figure 2. In all honesty I think the oral data are unnecessary for the authors to make their claims, but by showing the infectious titers one can see that these are very low compared to lung titers and thus probably do not contribute much to the disease. For what it is worth, I think infectious titers are infinitely more informative than RNA loads.
- 2) There is greater explanation of the animal model and the pros and cons. Admittedly much of this is in the rebuttal, but it helped in understanding the context in this rapidly changing field. If anything, more of this explanation needs to go into the manuscript.
- 3) The addition of the experiment to sequence viral genomes from treated animals and confirm the increase in substitutions provides evidence that MK-4482 is acting as an RNA mutagen. This provides important mechanistic insight.

This work is highly significant as the need for an oral drug to treat SARS-CoV-2 infections is critical. These data support the examination of MK-4482 in clinical trials.

Responses to individual Reviewers:

Reviewer 1:

Point 1. “It should be added to the discussion, however, that a recently released preprint reported considerably greater efficacy of MK-4482 against SARS-CoV-2 in Syrian golden hamsters, resulting in a reduction of RNA copies by approx. 3 log units and of lung virus titers by 3.3 log units”

Response to 1: *We thank the reviewer for ensuring that our study is current within this extremely fast moving field, and have added discussion of the preprint manuscript to the Discussion. Regarding the observed 1-log difference between the two studies, the preprint study differs in a number of key aspects that could account for this difference. These include differences in virus strains as well as in the titer of virus used for challenge. These parameters, as well as others have been shown to affect characteristics of the hamster model in other published studies (Imai et al. 2020. PNAS 117, 16586-16595; Chan et al. 2020 Clin. Infect. Dis 71, 2428-2446; Hou et al. Science 2020 370, 1464-1468). More importantly, the data presented in the preprint serves to further support the findings in our study. Other differences between the two studies are also discussed. **Lines 180 to 196.***

Point 2. “[In reference to the oral swab data] How can it be that this ratio [between RNA viral level and virus load based on TCID₅₀] is so strikingly different in the oral swabs, and how can these data be interpreted? These discrepancies need to be clarified...[P]ossibly a problem of the model or the sampling technique, the lack of correlation of RNA copies to infectious particles and the brief and inconsistent virus presence in the hamster upper respiratory tract confound interpretation of the effect on MK-4432 on virus shedding. Conclusions regarding the effect of treatment on viral transmission cannot be reached without an actual hamster transmission study. Possible limitations of the model need to be considered. This discussion should also cover the recently published data of efficacy of MK-4482 against SARS-CoV2 replication and spread in ferrets, since in ferrets MK-4482 reduced virus burden in the upper respiratory tract and blocked virus transmission”

Response to 2: *We appreciate the reviewer raising these points centered around the question of transmission within the hamster model. We now include a discussion raising the possible limitations of the oral shedding aspect of the study and how this relates to a possible effect of the drug on transmission – especially given the recent study in ferrets, which is more suited to assess the impact on transmission. The extremely ‘tight’ PCR results from the oral swabs in our study would indicate that the sampling technique is highly reproducible between animals and unlikely to be the problem. Whilst hamsters can be used to study transmission, the ferret is probably the better model to study this parameter. In the recent ferret transmission study, virus replication was shown to be localized to the upper respiratory tract, with neither virus or viral RNA being found in the lung. This combined with the extensive and historic use of the ferret to study transmission of other respiratory viruses (eg., flu) makes the ferret an ideal model for analysis of SARS-CoV-2 transmission. In the hamster, SARS-CoV-2 infection involves both lower as well as upper respiratory regions, with the high levels of virus in the lung combined with the associated lung pathology making the hamster suited to assessment of lung disease. **Lines 197 to 207.***

Point 3. “In lines 179-180, please update your statement that this “is the first demonstration of inhibition of SARS-CoV-2[...] on any animal model” by MK-4482, since there are published MK-4482 ferret data, as well as preprint releases covering use of MK-4482 in hamsters and mice”.

Response to 3: *We thank the reviewer for catching this item. At the time of our initial manuscript submission in early October, neither the hamster preprint study nor the ferret study had yet been submitted. This situation has changed in the interim, and we have brought the wording up to date. **Lines 180 to 181 & 197 to 199.***

Point 4. “The implied promise of synergistic effects of combination therapy with two nucleoside analogs, one an RNA mutagen and the other a chain terminator (lines 214-216), may be problematic, since in reality the latter drug undermines the MoA of the former”

Response to 4: *We agree that the efficacy of combinations of drugs would be difficult to predict and may prove problematic based on potential competing or interfering MoAs. We have clarified in the text that such combinations would need to be assessed in vitro and then in preclinical animal models to identify any adverse interactions. **Lines 248 to 250.***

Reviewer 3:

Point 1. “In all honesty I think the oral data are unnecessary for the authors to make their claims, but by showing the infectious titers one can see that these are very low compared to lung titers and thus probably do not contribute much to the disease. For what it is worth, I think infectious titers are infinitely more informative than RNA loads”

Response to 1: *We agree with the reviewer on the quantitation of virus titers (see also Response to Reviewer 1, Point 2).*

Reviewer #1 (Remarks to the Author):

The authors have fully addressed my remaining concerns.